Transcriptomic analyses reveal the potential regulators of the storage root skin color in sweet potato

Zhang Aicen
Yan Hui yanhui_sweetpotato@163.com
Tang Wei
http://orcid.org/0000-0001-6815-5461 Li Chen
Gao Tianqi
Song Weihan
Gao Runfei
Tang Wei
Kou Meng
Wang Xin
Zhang Yungang
Li Qiang instrong@163.com
Xuzhou Institute of Agricultural Sciences in Jiangsu Xuhuai District/Key Laboratory of Biology and Genetic Breeding of Sweet Potato, Ministry of Agriculture and Rural Affairs , Xuzhou , China
Irfan Mohammad
Electronic publication date: 2025 Dec 1
Publication date: 2025
Volume: 13
Electronic Location ID: e20231
Received 2025 May 26; Accepted 2025 Sep 23
Copyright: © 2025 Zhang et al.
Copyright year: 2025
Copyright holder: Zhang et al.
License: This is an open access article distributed under the terms of the Creative Commons Attribution License, which permits unrestricted use, distribution, reproduction and adaptation in any medium and for any purpose provided that it is properly attributed. For attribution, the original author(s), title, publication source (PeerJ) and either DOI or URL of the article must be cited.
License URL: https://creativecommons.org/licenses/by/4.0/

Keywords: Storage root skin color, Anthocyanin, Gene expression, MYBs, Sweet potato

Funding: CARS-10, Sweetpotato;Xuzhou Academy of Agricultural Sciences Research Fund Project JC2024001 National Natural Science Foundation of China 32301842 This work was supported by the earmarked fund for CARS-10, Sweetpotato; Xuzhou Academy of Agricultural Sciences Research Fund Project (JC2024001) and National Natural Science Foundation of China (32301842). The funders had no role in study design, data collection and analysis, decision to publish, or preparation of the manuscript.

==============================
Background

Sweet potato (Ipomoea batatas (L.) Lam.) is an important storage root crop exhibiting diverse storage root skin and flesh colors across varieties. The storage root skin color (SRSC) is a vital commercial trait which significantly influences the marketability and consumer preference of sweet potato.

Methods

To clarify the regulatory mechanism of SRSC, the sweet potato storage root of two clones derived from a common origin, designed as M1-125 (red skin and yellow flesh) and M1-125T (yellow skin and yellow flesh), were sampled and transcriptomic sequenced in this study.

Results

Comparative analysis revealed that differentially expressed genes (DEGs) in the root skin were predominantly enriched in flavonoid and anthocyanin biosynthesis pathways. Key structural genes, including chalcone synthase (CHS), chalcone-favanone isomerase (CHI), favanone 3-hydroxylase (F3H) and dihydroflavonol 4-reductase (DFR), exhibited higher expression levels in the root skin of M1-125. Notably, a subset of myeloblastosis (MYB) transcription factors showed significant expression changes between two genotypes. Based on the protein-protein interaction (PPI) network, phylogenetic analysis and expression pattern, IbMYB75, IbMYB3, IbMYB6 and IbMYB4 were thought to be the cooperative regulators of root skin color. These findings provide valuable insights and foundation for further elucidating the molecular mechanisms governing skin color of sweet potato storage root.

Introduction

Sweet potato (Ipomoea batatas (L.) Lam., 2n = 6x = 90) is a globally important economic crop, extensively cultivated in many countries such as India, South Africa and China (El Sheikha & Ray, 2017). In addition to be served as animal feed, industrial raw materials and energy resource, sweet potato has gained increasing attention as a nutritional powerhouse. Its edible components including storage roots, leaves and petioles, are rich sources of dietary fiber, vitamins, carotenoids, anthocyanins, etc. (Katayama et al., 2017; Tanaka et al., 2017; Yang et al., 2018), making them valuable for functional food processing. As a hexaploid species with a complex genome, sweet potato has gone through intricate processes of evolution and selection, hybridization and vegetative reproduction. These mechanisms have contributed to remarkable genetic diversity, reflected in extensive phenotypic variation in storage root size, shape, skin/flesh color, and flavor profiles (Katayama et al., 2017; Yang et al., 2020). Such diversity presents both opportunities and challenges for genetic improvement and molecular breeding programs.

The phenotypic diversity of sweet potato is particularly evident in its storage root coloration, which exhibits a broad spectrum ranging from white and yellow to deep orange and purple (Laveriano-Santos et al., 2022). Compared with lighter fleshed varieties, yellow- and orange-fleshed sweet potato possess higher content of carotenoids, while high levels of anthocyanins are detected in purple-fleshed varieties (Teow et al., 2007; Li et al., 2009; Tang, Cai & Xu, 2015). The anthocyanins does not only influence the appearance of storage root, but also enhance its nutritional value (Mohanraj & Sivasankar, 2014). Anthocyanins belong to flavonoids, a kind of secondary metabolite associated with stress resistance in plants (Gould, 2004). The anthocyanin biosynthesis pathway has been characterized in great detail across various plant species (He et al., 2020; Li, An & Wang, 2020; Li et al., 2021; Zhong et al., 2022; Do et al., 2023; Li et al., 2023a), which involved multiple conserved structural genes. This pathway initiates with the phenylpropanoid branch, including key enzymes such as phenylalanine ammonia lyase (PAL), cinnamate-4-hydroxylase (C4H) and 4-coumarate: coenzyme A ligase (4CL). Subsequent steps proceed through the flavonoid backbone formation catalyzed by chalcone synthase (CHS), chalcone-favanone isomerase (CHI), favanone 3-hydroxylase (F3H) in the formation of dihydroflavonols. The final modification and transport of anthocyanins are mediated by dihydroflavonol 4-reductase (DFR), anthocyanidin synthase (ANS), UDP-glucose flavonoid 3-O-glucosyltransferase (UFGT) and glutathione S-transferase (GST) (Sunil & Shetty, 2022; Do et al., 2023).

Based on the well-characterized anthocyanin biosynthetic pathway, recent studies have elucidated the crucial regulatory roles of various transcription factors (TFs) in modulating this metabolic process. Many TF families, like WRKY, NAC, ERF, MYB, bHLH, have been verified to work in anthocyanin biosynthesis through regulating structural genes (Zhou et al., 2016; Geng et al., 2022; Ni et al., 2023; Zhang et al., 2024). Among these regulators, R2R3-MYBs have emerged as particularly pivotal players, primarily functioning through the formation of MBW (MYB-bHLH-WD40) ternary protein complexes. It was reported RcMYB1 can interact with two bHLH proteins (RcBHLH42 and RcEGL1) and RcTTG1 (WD40 protein) to form two MBW complexes, both of which enhanced the anthocyanin accumulation in rose (Hou et al., 2023). Similarly, the strawberry FaMYB9 or FaMYB11 interacted with FabHLH3 and FaTTG1 to promote the proanthocyanidin biosynthesis (Schaart et al., 2013). In purple-fleshed sweet potato, IbMYB1 and some MYB1-like genes has been identified to make great contribution to activate the expression of anthocyanin-related genes (Mano et al., 2007; Zhang et al., 2021; Hou et al., 2023). The expression of IbWD40 gene showed a positive correlation with anthocyanin contents across various purple-fleshed sweet potato varieties (Dong et al., 2014). IbMYB1-2 initiated the transcription of structural genes in anthocyanin synthesis pathway along with its partner gene IbbHLH42 (Hou et al., 2023).

Up to now, many methods have been used for employing anthocyanins biosynthesis in sweet potato, including physiological and biochemical assays, molecular experiments and large-scale sequencing (Mano et al., 2007; Su et al., 2019; Zhang et al., 2020; Song et al., 2024). Among these, transcriptome sequencing (RNA-seq) technology is a powerful tool enabling comprehensive identification of differentially expressed genes (DEGs) associated with anthocyanin-mediated color variation across different sweet potato varieties (Dong et al., 2014; Zhang et al., 2021, 2022). Despite these advances, current research has predominantly focused on color variation of fresh, while the regulatory mechanisms governing SRSC remain comparatively understudied. In fact, sweet potato storage root skin color is an important appearance quality trait which directly affects the market acceptance, while elucidating its coloration mechanism can provide a theoretical foundation for breeding sweet potato cultivars with higher marketability. Weighted Gene Co-Expression Network Analysis (WGCNA) analysis between Sushu8 (with red skin and light orange flesh) and its mutant Zhengshu20 (with yellow skin and light orange flesh) helped to select a key module that enriched with DEGs highly related to the regulation of anthocyanin metabolism (Yang et al., 2020). Comparative transcriptomic analysis of Zheshu 81 (with red skin and yellow flesh) and its spontaneous mutation (with yellow skin and yellow flesh) revealed significant correlations between MYB transcription factor expression patterns and anthocyanin accumulation in root skin (Zhao et al., 2022). However, the specific regulatory factors associated with skin coloration remain to be further elucidated.

In the present study, we performed transcriptomic analysis for sweet potato line M1-125 (red skin and yellow flesh) and M1-125T (yellow skin and yellow flesh). Our analysis emphasized the important role of anthocyanin biosynthesis in root skin color regulation of sweet potato, and explored novel potential regulators, providing theoretical basis for understanding the SRSC regulation mechanism more detailed.

Materials and Methods

Plant materials and growing condition

The line M1-125 (red skin and yellow flesh) and line M1-125T (yellow skin and yellow flesh) were two clones derived from a hybrid offspring with ‘Xushu 45’ as the maternal parent. They were planted in the modern agricultural experimental farm of the Xuzhou sweetpotato research institute. The flesh and skin of their storage root were sampled with three biological replicates at 120 DAP (days after planting), then washed and cut into small pieces. All samples were immediately frozen in liquid nitrogen and subsequently stored at −80 °C for RNA extraction and library construction. In the second year, progenies of two materials were cultivated, and their leaves, stems, petioles, flesh, and root skin were sampled in 120 DAP following the aforementioned method for RNA extraction, which was subsequently used for RT-qPCR validation experiments.

RNA-seq libraries preparation and data analysis

Total RNA was prepared from flesh and root skin samples using TRIzol reagent kit (Invitrogen, Carlsbad, CA, USA). The concentration and integrity of RNA were measured by an Agilent 2100 bioanalyzer (Agilent Technologies, Palo Alto, CA, USA) and RNase-free agarose gel electrophoresis, respectively. The mRNA was enriched by Oligo (dT) beads. The completed mRNA was then used for library construction following the manufacturer’s protocol of NEBNext Ultra RNA Library Prep Kit for Illumina (NEB #7530; New England Biolabs, Ipswich, MA, USA). All mRNA-seq libraries were sequenced on Illumina NovaSeq 6000 platform in Gene Denovo Biotechnology Company, Guangzhou, China. The statistical power of this experimental design, calculated in RNASeqPower is 0.88.

Fastp (v0.21.0) (Chen et al., 2018) (—length_required = 50) was used for removing adapter and low-quality reads of the raw sequencing data. The filtered clean reads were then mapped to the Beauregard (v3) reference genome (http://sweetpotato.uga.edu/sweetgains_Beauregard_v3_asm_anno.shtml) using HISAT2 (v2.0.5) with default parameters (Kim, Langmead & Salzberg, 2015). The RNA-seq reads within per gene were counted using the featureCounts (Liao, Smyth & Shi, 2014) software, with the multiple align reads filtered out.

The counts matrix of all samples was used as the input of R package DESeq2 (v1.40.2) (Love, Huber & Anders, 2014) for the identification of DEGs. Genes with padj < 0.05 and |log2(fold change)| > 1 were confirmed as DEGs.

Gene Ontology and Kyoto Encyclopedia of Genes and Genomes enrichment analysis

First all protein sequences of the reference genome were submitted to the online website (http://eggnog5.embl.de/#/app/home) to blast against all collected plant proteins in EggNOG Database (Hernández-Plaza et al., 2023). Then the python script parse_eggNOG.py (https://github.com/Hua-CM/HuaSmallTools.git) was used to match the genes with Gene Ontology (GO) terms/Kyoto Encyclopedia of Genes and Genomes (KEGG) pathway one by one. The selected gene list and annotated files generated by EggNOG were regarded as input of the R package clusterProfiler, the p value was adjusted by its inside “BH (Benjamini & Hochberg)” method. Pathways with FDR values less than 0.05 after correction were considered to be significantly enriched.

Annotation of structural genes and TFs

The annotation and name of structural genes and TFs was the integrated results from plantTFDB (https://planttfdb.gao-lab.org/) (He et al., 2010) and STRING database (https://cn.string-db.org/) (Szklarczyk et al., 2023). The protein sequences of target genes were submitted to the online website of both databases. The TF family and their homologous protein with best hit in Arabidopsis thaliana were given by plantTFDB, while STRING database provided more detailed functional annotation of proteins.

Reverse transcription quantitative real-time PCR assay

Total RNA was extracted from ground powder of five different tissues (line M1-125 and M1-125T were included) using the RNApure Plant Kit (DNase I) (CWBIO, Beijing, China). Agilent 2100 bioanalyzer (Agilent Technologies, Palo Alto, CA, USA) was used for purity and concentration assessment of the total RNA, and agarose gel electrophoresis was performed to check their integrity. Then the first-stranded cDNAs were synthesized by using SuperScript II Kit (TaKaRa, Shiga, Japan) in terms of the manufacturer’s instructions. All samples of cDNA were stored under −20 °C before using. RT-qPCR was conducted using the SYBR reagent with PCR programs as below: 95 °C for 60 s, 40 cycles with 15 s at 95 °C and 15 s at 60 °C per cycle, 20 s at 72 °C for elongation. Each sample was biologically and technically replicated for three times. The relative expression levels were calculated using the 2−∆∆CT method (Livak & Schmittgen, 2001) with sweet potato tublin gene referred as an internal control (Song et al., 2022). The CDS sequences of candidate genes, which were used for primer design can be searched in the specific website of sweet potato (https://sweetpotato.uga.edu/) by inputting gene ID. The gene expression level in leaf of M1-125T was regarded as ‘1’ to reflect the relative changes of gene expression in each tissue. The primers were designed by using Primer3.0 (https://www.primer3plus.com/) with the product size ranging from 80–200 bp. All PCR primers are listed in Table S8.

Results

Sampling of plant material and mapping of RNA-seq data

To investigate the regulation mechanism underlying skin color variation in sweet potato storage roots, the skin (S) and flesh (F) of line M1-125 (red skin and yellow flesh) and line M1-125T (yellow skin and yellow flesh) were sampled (Fig. 1A), with three biological replicates for each material. When the gene expression dynamics were characterized between the no color-changed flesh and the color-changed skin, focusing on the pathways specifically enriched or genes highly expressed in root skin will help to identify key regulators for the skin coloration. After RNA-seq library construction and sequencing, approximately 615.6M raw reads were obtained, of which 99.01–99.53% were clean reads (Table S1).

Figure 1 Phenotype and RNA-seq data analysis of line M1-125 and M1-125T.

(A) Samples of M1-125 (left) and M1-125T (right). Colors of storage root skin and flesh of M1-125 (red skin and yellow flesh) and M1-125T (yellow skin and yellow flesh) were shown. (B) Heatmap showing the correlation between different samples, the number indicated the Pearson correlation coefficient (PCC). (C) Principal component analysis (PCA) for the 12 samples. (D) Violin plot showing the average gene expression level for flesh and skin in M1-125 and M1-125T. The number of expressed genes (FPKM > 0.5) were labelled. Wilcoxon rank sum test, “ns.” indicated no significant difference.

The clean reads were then mapped to the Beauregard (v3) reference genome. Subsequently, the aligned read counts for each protein-encoded gene with high confidence (n = 225,112) was quantified and normalized by sequence depth and gene length to obtain the FPKM values (Fragment Per Kilobase of transcript per Million mapped reads), which represented the expression levels of genes. Based on the FPKM values, we used the cor function of R to calculate the correlation of the 12 samples. The results showed the biological replicates from each material were well correlated with the pearson correlation coefficient (PCC) over 0.99, while the gene expression of skin and flesh exhibited more differences (PCC < 0.42), and the PCC between M1-125 and M1-125T ranged from 0.94 to 0.97, suggesting their gene expression difference in storage root (Fig. 1B). In consistent with the result of correlation analysis, principal component analysis (PCA) showed the 12 samples could be clustered into four groups, notably, biological replicates were grouped more tightly than different tissue types or genotypes (Fig. 1C). To generate an overview of expression patterns across the whole genome, we next calculated the average expression levels and counted the number of expressed genes (FPKM > 0.5) for the four groups of materials. As shown in Fig. 1D, there was no significant expression difference between flesh and skin or M1-125 and M1-125T, respectively. Totally 64,068–67,417 genes were expressed in storage root, while more than half of the protein-encoded genes remained transcriptionally inactive. Collectively, the above results confirm the high quality and reproducibility of our RNA-seq data, supporting their suitability for downstream analyses.

Identification of differentially expressed genes

The differentially expressed genes (DEGs) between M1-125 and M1-125T were identified by using DESeq2 with the cutoff set as padj < 0.05 and log2(fold change) > 1 (Figs. 2A, 2B, Table S2). There were 2,833 and 2,996 DEG in two comparison groups (flesh and skin), respectively (Figs. 2A, 2B). Compared with the M1-125, 831 genes were down-regulated and 2002 genes were up-regulated in flesh of M1-125T (Fig. 2A), on the contrary, there were more down-regulated genes (n = 1,734) than up-regulated ones (n = 1,262) (Fig. 2B) in the skin of M1-125T. Given the established role of transcription factors (TFs), particularly MYBs and bHLHs, in root color control of sweet potato through anthocyanin accumulation (Li et al., 2022), we then specifically focused on the expression changes of TFs, as expected, we detected 57 down-regulated and 172 up-regulated TFs in flesh of M1-125T relative to M1-125, among which ERF, bHLH and NAC TFs had the most numbers (Fig. 2C), moreover, 103 down-regulated and 63 up-regulated TFs were found in skin sample between two materials, with bHLH, MYB and WRKY TFs having the highest number (Fig. 2D). However, these TFs showed opposite trend of change in flesh and skin, the majority of bHLHs and MYBs were up-regulated while most of them were down-regulated in M1-125T (Figs. 2C, 2D, Table S3). These tissue-specific transcriptional changes maybe the basis of the observed skin color variation.

Figure 2 DEGs between line M1-125 and M1-125T.

(A) Volcano plot showing the DEGs between the flesh of M1-125 and M1-125T. The blue dots represent the down-regulated genes, the yellow dots represent the up-regulated gene, and the grey dots represent genes with no expression differences. (B) Volcano plot showing the DEGs between the root skin of M1-125 and M1-125T. The blue dots represent the down-regulated genes, the yellow dots represent the up-regulated genes, and the grey dots represent genes with no expression differences. (C) Numbers of differentially expressed TFs between the flesh of M1-125 and M1-125T. The yellow bars represent the up-regulated TFs, the green bars represent the down-regulated TFs. (D) Numbers of differentially expressed TFs between the root skin of M1-125 and M1-125T. The yellow bars represent the up-regulated TFs, the green bars represent the down-regulated TFs.

GO and KEGG enrichment analysis of DEGs

To explore if the DEGs were involved in some function pathways related to the color-related regulation, we first performed GO enrichment for these DEGs with all annotated genes as background. Here, the top 20 GO terms with the highest significance were displayed, notably, the down-regulated genes in M1-125T-S vs M1-125-S were enriched in completely different biological processes as compared with other three groups (M1-125T-F vs M1-125-F up, M1-125T-F vs M1-125-F down, M1-125T-S vs M1-125-S up) (Fig. 3, Table S4). We found the other three groups were all associated with plant hormone pathways, such as ‘regulation of jasmonic acid mediated signaling pathway’ (M1-125T-F vs M1-125-F up, Fig. 3A), ‘cytokinin biosynthetic process’ (M1-125T-F vs M1-125-F down, Fig. 3B), and ‘negative regulation of cytokinin-activated signaling pathway’, ‘jasmonic acid hydrolase’, ‘jasmonic acid biosynthetic/metabolic process’ (M1-125T-S vs M1-125-S up, Fig. 3C). However, it is worth noting that M1-125T-S vs M1-125-S down-regulated genes were overrepresented in ‘flavonoid biosynthetic process’, ‘anthocyanin-containing compound biosynthetic/metabolic process’ (M1-125T-S vs M1-125-S up, Fig. 3D), indicating a suppression of anthocyanin-related pathway in the root skin of M1-125T.

Figure 3 GO enrichment of DEGs.

The top 20 enriched GO terms of up-regulated genes (A) and down-regulated genes (B) in flesh, as well as up-regulated genes (C) and down-regulated genes (D) in root skin, respectively. The color indicated the −log10(FDR) value, Fisher test, GO terms with FDR < 0.05 were considered to be significantly enriched.

We also conducted KEGG enrichment analysis for the same gene set, the ‘Thiamine metabolism’ and ‘Linoleic acid metabolism’ pathway had the most significant enrichment in flesh (Fig. S1, Table S5), while ‘flavonoid biosynthesis’ pathway was the most enriched in M1-125T-S vs M1-125-S down-regulated genes, corroborating the GO enrichment findings. These results collectively highlight the crucial role of anthocyanin biosynthesis pathway in mediating SRSC variation.

Expression profiles of structural genes related to color regulation and their interaction network

To shrink the scope of genes that cause root skin color mutation and clarify their interaction relationship, firstly, a hierarchical clustering analysis was conducted for all merged DEGs (n = 5,512) from no color-changed flesh and color-changed root skin. The DEGs were finally groups into six clusters, genes in cluster 2 (n = 533) and cluster 5 (n = 807) were more expressed in root skin and flesh, respectively; genes in cluster 3 (n = 765) and cluster 6 (n = 1,281) had the highest expression level in the flesh of M1-125 and M1-125T, respectively (Fig. 4A). What is more, there was a specific cluster with genes highly expressed in the root skin of M1-125 (cluster 3, n = 1,235) (Fig. 4A), both GO and KEGG enrichment analysis showed the biological process ‘flavonoid biosynthesis’ was most relevant to cluster 3 (Fig. 4B), indicating cluster 3 is a key module related to the SRSC. We then extract genes located in flavonoid/ anthocyanidin biosynthesis pathway in cluster 3, totally 63 structural genes were selected, including one PAL gene, five 4CL genes and seven CCOAOMT genes in phenylpropanoid pathway; nine CHS genes, nine CHI genes, six F3H genes, five F3′H and four FLS genes in early biosynthesis pathway; as well as 10 DFR genes, four UFGT genes, three GST genes in late biosynthesis pathway (Fig. 4C, Table S6). The majority of these genes were not expressed in the flesh, but exhibited high expression levels in the root skin of M1-125, especially the GST, F3H, FLS and some of the CHS genes (Fig. 4C). To further detect the key proteins that interacted with these structural genes, all protein sequences of genes in cluster 3 were submitted to the online STRING database for the construction of PPI network (protein-protein interaction network), and the subnetwork with structural genes as core part were extracted and shown in this study (Fig. 4D). The CHS, DFRA proteins were predicted to have most interaction relationship with other proteins, furthermore, these structural enzymes formed core connections with multiple regulatory proteins, including key MYB TFs (MYB3/4/6/12/42/75), bHLH proteins (TT8) and several WRKY TFs (WRKY33/40/70) (Fig. 4D). These interactions suggest a coordinated regulatory mechanism for controlling anthocyanin biosynthesis and SRSC.

Figure 4 Screening of structural genes in anthocyanin biosynthesis pathway and their PPI network.

(A) Hierarchical clustering of all DEGs from flesh and root skin. The color indicates the expression levels. (B) GO and KEGG enrichment of cluster 3 genes in (A). Fisher test, pathways with FDR < 0.05 were considered to be significantly enriched. The color indicates the −log10(FDR) value. (C) Expression profiles of structural genes in anthocyanin biosynthesis pathway. The color indicates the log2(FPKM + 1) value. All biological replicates were shown. (D) PPI network (protein-protein interaction network) of structural genes. The color and the circle size represents the number of interaction edges for each protein. The thickness of the lines represents the predicted possibility of interaction relationship.

Carotenoid metabolism is another crucial pathway involved in color regulation. Although we did not find significant enrichment of DEGs in this pathway, considering the presence of yellow skin, here we also examined the expression profiles of genes located in the carotenoid biosynthesis pathway. Accordingly, seven DEGs were found to be located in this pathway, among which five of them were NCEDs, the other two were CYP97C and CCD, respectively (Fig. S2). The NCEDs showed higher expression levels in yellow flesh of both M1-125 and M1-125T, and four NCEDs were up-regulated in M1-125T-S as compared to that in M1-125-S, while CYP97C and CCD were more expressed in M1-125-S (Fig. S2). Overall, the number of DEGs involved in carotenoid regulation was considerably lower than that of anthocyanin-related genes, suggesting a major role of anthocyanins in SRSC variation from red to yellow.

Selection of candidate MYBs related to SRSC regulation of sweet potato

Considering the important role of MYBs in color regulation among different plants and their widespread presence in the above-mentioned PPI network, we then paid attention to the selection of MYBs potentially involved in sweet potato root skin coloration. We collected all differentially expressed MYBs (n = 26) in flesh and root skin, based on their expression dynamics, all MYBs were categorized into four groups (Fig. 5A). Cluster 2 were found to contain nine MYBs showing significant high expression in the root skin of M1-125 (Fig. 5A), among which IbMYB75 and IbMYB6 were found to exhibited higher expression fold change in root skin between M1-125 and M1-125T, by contrast, they were lowly expressed in the yellow flesh of both M1-125 and M1-125T (Fig. 5A). Additionally, IbMYB3/4/12 and two MYB-like TFs were also specially expressed in M1-125 skin.

Figure 5 Expression and neighbor joining tree of differentially expressed MYBs between M1-125T and M1-125.

(A) Heatmap showing the expression changes of MYBs in flesh and root skin. The left heatmap indicates the expression level, while the right heatmap represents the fold change. DEGs (padj < 0.05, |log2fold change| > 1) were labeled by ‘*’, and the no-expressed genes were labeled by ‘NA’. (B) Hierarchical cluster analysis of MYBs based on the protein sequence percentage identity of the MYBs. The dots with different color represent MYBs come from different plant species.

To learn more about the possible function of these MYBs, we first characterized their conserved domains and motifs. Sequence analysis revealed that all identified MYBs belong to the R2R3-MYB subfamily, containing two characteristic Myb_DNA-binding domains (PF00249) (Fig. S3A). We identified 15 distinct motifs, with Cluster 3/4 MYBs exhibiting the highest motif diversity. Notably, motif 1~motif 3 were universally conserved across all MYBs, while showing the highest degree of sequence variation (Fig. S3B), suggesting their potential role in functional diversification among different MYB proteins. In the M1-125-S highly expressed Cluster 2, IbMYB12 uniquely contained motif 7 in addition to the core motifs, whereas IbMYB-like possessed motifs 11 and 12, and IbMYB3/4 contained motif 4. Next we collected a set of MYBs previously reported color-related MYBs from various plant species (Table S7) (Wang et al., 2020a), based on the all vs all global alignment of the protein sequences of 68 MYBs, including the differentially expressed MYBs shown in Fig. 5A, we conducted a hierarchical clustering analysis according to the sequence percentage identity. The clustering results revealed that the two IbMYB75 genes (Ibat.Brg_v3.12AG004780, Ibat.Brg_v3.12AG004740) were clustered with anthocyanin biosynthesis activators, including AtMYB75, AtMYB90, AtMYB113, VvMYBA1 and MYB10 from different plants (Fig. 5B). IbMYB6 (Ibat.Brg_v3.07FG007500, Ibat.Brg_v3.07AG005840), IbMYB3 (Ibat.Brg_v3.04FG026450) and IbMYB4 (Ibat.Brg_v3.09FG001230) were clustered with anthocyanin biosynthesis suppressors, such as VvMYB6, PavMYB111, MdMYB111 and FaMYB1/FcMYB1, suggesting IbMYB75/3/4/6 may function coordinately in regulating sweet potato SRSC through anthocyanin biosynthesis pathways.

To validate the expression patterns and heritability of these MYB genes, we performed RT-qPCR analysis using samples from different tissues from the progeny of M1-125 and M1-125T (Fig. 6). The results demonstrated that all four-type MYB genes exhibited relatively low expression levels in aerial tissues but were highly expressed in root skin of M1-125. As expected, IbMYB75 and IbMYB6 showed more significant expression difference between M1-125 and M1-125T skin. These findings suggest that the MYB-mediated regulatory mechanism underlying skin coloration can be stably inherited through vegetative propagation.

Figure 6 RT- qPCR analysis of IbMYB3, IbMYB4, IbMYB6 and IbMYB75 across different tissues of progeny of M1-125 and M1-125T.

Statistical analyses were performed using three biological replicates, with data presented as mean ± standard deviation (SD). ANOVA test, different letters on the bar indicate significant differences at the p < 0.05 level.

Discussion

The SRSC can affect the appearance of sweet potato, thereby influence the acceptance of consumers (Yang et al., 2020). But previous studies paid little attention to the mechanism of SRSC regulation in sweet potato. Recent advances in transcriptomic sequencing technologies have emerged as powerful tools for identifying candidate genes involved in specialized metabolic pathways (Zhang et al., 2022; Tao et al., 2013). Moreover, the availability of high-quality hexaploid reference genomes, including ‘Taizhong6’ (Yang et al., 2017), ‘Xushu18’ (https://plantgarden.jp/ja/list/t4120/genome/t4120.G001) and ‘Beauregard (v3, SweetGAINS Project)’, has significantly enhanced our capacity for functional genomic studies in sweet potato. In this study, we conducted comparative transcriptomic analysis of red-skinned sweet potato and their yellow-skinned mutants using the Beauregard (v3) genome as reference. Our analysis encompassed approximately 66,000 expressed genes for comprehensive downstream evaluation (Fig. 1D). Based on the gene expression pattern, two color regulatory pathways were evaluated, including the carotenoid and anthocyanin biosynthesis pathways (Figs. 4C, S2). We found the majority of the carotenoid-related DEGs (5 of 7) were NCEDs, which have been reported to play important role in the regulation of carotene components (Sun et al., 2012; Jia et al., 2022; Lei et al., 2022), indicating a possible influence of carotenoid on the color variation of SRSC. However, most changes were observed in anthocyanin/flavonoid-related pathways, in which the number of DEGs was far more than that of carotenoid-related pathway. The distinct expression profiles observed between no color-changed flesh and color-changed skin not only illustrated the importance of anthocyanin biosynthesis for the skin color, but also revealed tissue-specific regulatory mechanism in storage roots. Therefore, we speculate the variation color variation from dark (red) to light (yellow) of SRSC was mainly caused by the downregulation of anthocyanin pathway, while the contribution of the carotenoid pathway and whether there is synergistic effect between the two pathways remains to be further evaluated. It will be more interesting to explore if there are common upstream regulatory factors between the two pathways. Anyhow, our findings provide theoretical and data basis into the spatial regulation of anthocyanin accumulation in sweet potato roots.

TFs serve as crucial regulators regulatory factors during plant growth and development (Li et al., 2019; Xie et al., 2019; Diao et al., 2020), which also play important roles in modulating anthocyanin accumulation (Li et al., 2022, 2023b). Emerging evidence from various species highlights the diverse TF families involved in this process. For example, the overexpression of PpNAC25 in peach improved its anthocyanin content by promoting the expression levels of anthocyanin biosynthesis and transport genes (Geng et al., 2022). The VvWRKY5-VvMYBA1 interaction activates VvUFGT in grape (Zhang et al., 2024). CmbHLH16 maintained the anthocyanin homeostasis in petals of chrysanthemum under different light conditions (Zhou et al., 2022). MdERF3 positively regulates anthocyanin biosynthesis in apple (Li et al., 2023a). In our analysis, we found more down-regulated TFs in root skin of M1-125T, especially MYBs, ERFs and bHLHs (Figs. 2C, 2D). And the cluster of DEGs together with prediction of PPI network demonstrated the co-expression and interaction relationship between TFs and structural genes (Figs. 4A, 4D). Multiple structural genes were highly expressed in root skin of M1-125 and interacted with MYBs and bHLHs (TT8) (Fig. 4D), these findings provide candidate members of the MBW complex, and raise important questions about the mechanisms governing tissue-specific expression patterns between skin and flesh tissues, which warrant further investigation.

Considering the vital function of MYBs in color definition of purple-fleshed sweet potato (Mano et al., 2007; Kim et al., 2010; Dong et al., 2022; Hou et al., 2023) and their prominent representation in our protein-protein interaction (PPI) network, here the expression profile and cluster analysis of sequence similarity with MYBs from other species were studied (Fig. 5). Among them, IbMYB75, IbMYB3, IbMYB4 and IbMYB6 was lowly expressed in yellow flesh and exhibited high fold change in root skin, the clustering of sequence percentage identity revealed IbMYB75 was more correlate with AtMYB75 (AtPAP1), AtMYB90, AtMYB113 and VvMYBA1. AtMYB75, AtMYB90 and AtMYB113 are canonical upstream regulatory factors of the anthocyanin biosynthesis pathway in Arabidopsis (Muñoz-Gómez et al., 2021); VvMYBA1 is a determinant of color variation in grape (Lijavetzky et al., 2006). Positive regulatory factors MYB10 from different plant species (Wang et al., 2020a) were also be clustered into a bigger cluster with IbMYB75. In contrast, IbMYB6 was more correlate with VvMYB6, MdMYB111 and PavMYB111, while IbMYB3 and IbMYB4 were in the same cluster with AtMYB3, AtMYB4 and PavMYBR, etc. Moreover, the three type MYBs were clustered with MYB1 of strawberry in a bigger clade, while VvMYB6, MdMYB111 and FaMYB1/FcMYB1, AtMYB3, AtMYB4 were reported as inhibitors of anthocyanin synthesis (Wang et al., 2020a; Aharoni et al., 2001; Salvatierra et al., 2013). Intriguingly, these repressor-type MYBs also displayed elevated expression in red-skinned sweet potato, suggesting potential competitive binding between IbMYB3/4/6 and the activator IbMYB75. In fact, this competitive mechanism is an effective way for plants to maintain anthocyanin homeostasis. For instance, in peony flowers, PsMYB308 competently binds to PsbHLH1-3 with PsMYBPA2 to fine-tune the regulatory network and prevent excessive accumulation of anthocyanins in spotted areas (Luan, Tao & Zhao, 2024). VvMYB3 can inhibit anthocyanin biosynthesis by competitively binding VvMYC1 with VvMYBA1 in grape (Qin et al., 2025). AtMYB4 interacts with bHLH transcription factors TT8, GL3, and EGL3, thereby interfering with the transcriptional activity of the MBW complex (Wang et al., 2020b). Anyway, our findings provide insights into the transcriptional regulation of sweetpotato SRSC. However, further functional studies are needed to validate the roles of these candidate MYBs and their interplay in the anthocyanin biosynthesis pathway.

Conclusions

In this study, comparative transcriptome analysis of sweetpotato red-skinned line M1-125 and yellow-skinned line M1-125T was conducted to explored the storage root skin color (SRSC) regulation mechanism. We generated the genome-wide gene expression profile of storage root flesh and root skin of sweet potato, provided the detailed information related to flavonoid/anthocyanin biosynthesis pathways, as well as classical structure genes like CHS, CHI, F3H, and DFR based on the hexaploid genome. Moreover, TF families including ERF, bHLH, NAC, MYB and WRKY, were found to play important roles in SRSC changes. A subset of particular MYB transcription factors, including activator IbMYB75, suppressor IbMYB3, IbMYB6, and IbMYB4, collaboratively regulate SRSC. Our findings provide novel insights into the genetic regulation of sweetpotato SRSC, raise important questions about the mechanisms governing tissue-specific expression patterns between skin and flesh tissues that warrant further investigation, and lay a foundation for future genetic and biochemical studies aimed at modifying color traits in sweet potato breeding programs.

Supplemental Information

Supplemental Information 1 Summary of RNA-seq data.

Supplemental Information 2 List of DEGs between M1-125T and M1-125.

Supplemental Information 3 List of differentially expressed TFs between M1-125T and M1-125.

Supplemental Information 4 Enrichment of GO terms of DEGs between M1-125 T and M1-125.

Supplemental Information 5 Enrichment of KEGG pathways of DEGs between M1-125 T and M1-125.

Supplemental Information 6 Expression pattern and annotation of genes involved in anthocyanin biosynthesis pathway.

Supplemental Information 7 Sequence of protein for construction of phylogenetic tree.

Supplemental Information 8 Sequence of primers used for qRT-PCR.

Supplemental Information 9 KEGG enrichment of DEGs.

The enriched KEGG pathway of up-regulated genes (A) and down-regulated genes (B) in flesh, as well as down-regulated genes (C) in root skin, respectively. The color indicated the −log10(FDR) value, pathways with FDR < 0.05 were considered to be significantly enriched.

Supplemental Information 10 Expression profiles of structual genes in carotenoids biosynthesis pathway.

Supplemental Information 11 Conserved domain and motif analysis of different expressed MYBs.

Supplemental Information 12 R code and files for DESeq2 package.

Supplemental Information 13 R code and files for clusterprofile package.

Supplemental Information 14 MIQE checklist.

Supplemental Information 15 Raw data of qRT-pcr.

Additional Information and Declarations

Competing Interests

The authors declare that they have no competing interests.

Author Contributions

Aicen Zhang analyzed the data, prepared figures and/or tables, and approved the final draft.

Hui Yan conceived and designed the experiments, authored or reviewed drafts of the article, and approved the final draft.

Wei Tang performed the experiments, prepared figures and/or tables, and approved the final draft.

Chen Li performed the experiments, prepared figures and/or tables, and approved the final draft.

Tianqi Gao performed the experiments, prepared figures and/or tables, and approved the final draft.

Weihan Song analyzed the data, prepared figures and/or tables, and approved the final draft.

Runfei Gao analyzed the data, authored or reviewed drafts of the article, and approved the final draft.

Wei Tang analyzed the data, prepared figures and/or tables, and approved the final draft.

Meng Kou analyzed the data, authored or reviewed drafts of the article, and approved the final draft.

Xin Wang performed the experiments, authored or reviewed drafts of the article, and approved the final draft.

Yungang Zhang conceived and designed the experiments, authored or reviewed drafts of the article, and approved the final draft.

Qiang Li conceived and designed the experiments, authored or reviewed drafts of the article, and approved the final draft.

Data Availability

The following information was supplied regarding data availability:

The RNA-seq data is available at GEO: GSE294595.

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
