# Peer review of "Transcriptomic analyses reveal the potential regulators of the storage root skin color in sweet potato"

_PeerJ, doi:10.7717/peerj.20231_

## Round 0.1 · original submission · Major Revisions

· Academic Editor

Major Revisions

The manuscript was evaluated by two independent reviewers. Both found the work interesting but raised several concerns that must be addressed. The primary issue is related to the genetic background of the WT and mutant lines. The authors should clearly clarify the genotypes used in the study, supported by appropriate references. In addition to this, I have the following comments that should be addressed to improve the clarity and quality of the manuscript:

1) The authors primarily focus on anthocyanin biosynthesis genes and associated MYB transcription factors. However, considering the presence of a yellow-skinned variety, it is important to also examine the expression of carotenoid biosynthesis genes. Rather than biasing towards anthocyanins, I recommend reanalyzing the transcriptome data to include both anthocyanin and carotenoid metabolic pathways and revising the Results and Discussion sections accordingly.

2) Statistical Analysis: The statistical analysis throughout the manuscript is unclear. Please provide detailed information in the figure legends, including:
The type and number of biological and technical replicates used,
The type of statistical tests performed,
P-values,
And the criteria used to define significant fold changes, where applicable.

Addressing these points will significantly improve the rigor and interpretability of the study.

Reviewer 1 ·

Basic reporting

The authors conducted an interesting research on the skin color of sweetpotato and revealed a valuable insight. However, some of the typos should be addressed, such as in line 140 and so on. Also, some expressions might require a minor focus, such as in line 81, 'does' might be a better expression than 'can' in this sentence, and so on.

Experimental design

Serious concern is raised as follows,

1. Reviewer failed to notice why M1-125 and MT are wild type and natural mutant, which most of the sweetpotato originated from either cultivated or landrace. Authors should provide the reason to believe M1-125 is a wild type.

2. TaKaRa is a Japanese company; you need to check this kind of mistake.

3. The version of the genome (of course, it is v4 now) is available on the condition that the authors may not use this genome for identifying the ‘gene family’. Authors should have either the approval of Dr. Zhangjun Fei for those or change the genome version.

Validity of the findings

-

Reviewer 2 ·

Basic reporting

The manuscript is clearly written. The authors present their data and results logically and coherently, making the manuscript easy to follow. Figures and tables are well-designed, properly labeled, and relevant to the study’s objectives. Raw data are reported and appear to be provided in accordance with the journal’s data availability requirements.

Experimental design

The experimental design is appropriate for the study’s objective of comparing two sweet potato lines—M1-125 (wild type; red skin and yellow flesh) and its natural mutant (yellow skin and yellow flesh). The approach is relevant and well-structured.

However, the primer design methodology and statistical analysis procedures are not described in sufficient detail in the Methods section. These aspects should be clarified to ensure reproducibility and transparency.

Validity of the findings

The findings align with the research objectives and are supported by the raw data provided. The results are robust, and the authors’ interpretations are logical and consistent with the evidence presented.

Additional comments

The manuscript is well-prepared, and the obtained results are convincing and clearly presented. However, I have some suggestions and minor remarks that need to be addressed in the revised version before publication. My recommendation is Minor Revisions.

I. Introduction
The background provided is adequate and contextualizes the study well. The anthocyanin biosynthetic pathway involves a well-characterized sequence of enzyme-mediated reactions, each regulated by specific genes. The anthocyanin biosynthesis was described in detail with key references relevant to its biosynthesis pathway. However, it would strengthen the Introduction if the authors include a recent literature [https://www.mdpi.com/1422-0067/24/16/12946], which could be referenced in paragraph 2 to enhance the discussion of regulatory genes and pathway mechanisms.

II. Methods:
1. What is the harvesting time of sweetpotatoes, MT and WT? Was their harvesting time the same as the sampling time for analysis (120 DAP)?
2. Why did the authors not collect other sample tissues (leaves, stems, petioles, flesh, and root skin for qRT-PCR) at the same time that their storage roots were sampled (for RNA-seq)?. Herein, other sample tissues (leaves, stems, petioles, flesh, and root skin) were sampled in the second year.
3. Anyway, for mRNA-seq library construction, the mRNA was enriched using Oligo (dT) beads. Does it require treatment for the removal of rRNA? How about tRNA?
4. Please provide the access link of “the online website of both databases” (Line 181) and present them in the main text of the manuscript.
5. Noted that qRT-PCR primers are not listed in Table S8. It was listed in Table S7. Please revise.
6. Please provide the primer sequences of the reference gene (tublin) and add them to Table S7. In addition, provide the accession number/GenBank of these MYB genes (MYB3, 4, 6, 75) and tubulin, based on which the primer sequences were designed. Moreover, how these primer sequences were designed should also be described. Including this information for Materials and Methods to allow replication and building on the authors’ obtained results.

III. Results
1. Please specify “3 genes were submitted to the online STRING database for the construction of the PPI network” (L282-283).
2. Noted that “color-related MYBs from various plant species (Table S7)” (L310) were not found in Table S7. It was listed in Table S8. Please double-check and re-cite the data.
3. Similarly, double-check the citation data of Fig. 5A (L312) in paragraph 2 in sub-section “Selection of candidate MYBs related to SRSC regulation of sweetpotato”.
4. Significant expression difference between WT and MT tissues requires statistical analysis of these data (Fig. 6).
5. Revise: “n = 1260” (L232) → n = 1262

IV. Minor remarks:
- The following sentence needs to be rewritten to improve clarity: “To some extent, the same flesh color of WT and MT can be regarded as a control which helps to define regulators for the skin color.” (L203-204).
- Italicize the scientific name of the plant: Ipomoea (L63)
- Revise: “straw berry” (L103) → strawberry
- “differ e ntially” (Fig.2’s legend)
I have marked (text highlighted in yellow) some of the above minor remarks on the manuscript. Please use it for easy tracking.

With regards,

Annotated reviews are not available for download in order to protect the identity of reviewers who chose to remain anonymous.

---

## Round 0.2 · Minor Revisions

· Academic Editor

Minor Revisions

Thanks for revising the manuscript however there are a few minor concerns pointed out by reviewer 1.

Reviewer 1 ·

Basic reporting

The authors addressed the most of the reviewer's comment yet a concern bothers this reviewer as following.

I understood the origin of your lines. Yet, the information provides they are 'cultivars' not 'wild-type'. Also, M1-215T maybe denoted as 'mutant' yet is potential without solid evidences (e.g. it can be epigenetic if the heredity is not stable for several generations. For clarity, please revise the words into 'two clones originated from an origin' not 'wild-type' until those lines were found in 'isolated region without human contact for enough time'.

Experimental design

no comment

Validity of the findings

no comment

Additional comments

no comment

Reviewer 2 ·

Basic reporting

As commented in round 1

Experimental design

As commented in round 1

Validity of the findings

As commented in round 1

Additional comments

Dear authors,

Many thanks for the revisions to your manuscript. The authors have made efforts to improve the manuscript. The revised version has been greatly enhanced. Other than my comments and the authors’ responses, I have also read the Editor's and 1st reviewer’s comments and authors' responses to him/her as well. I have agreed with the reviewer's comments and sacrificed with the authors’ responses and changes that the author has made. I recommend this work for publication.

With regards,

---

## Round 0.3 · accepted · Accept

· Academic Editor

Accept

Authors have addressed all the comments raised during the review process therefore article can be accepted for publication it its current form.